# Design of Miniature Ultrawideband Active Magnetic Field Probe Using Integrated Design Idea

**DOI:** 10.3390/s23136170

**Published:** 2023-07-05

**Authors:** Yang Zhou, Zhaowen Yan, Zhangqiang Ma, Yang Zhao, Jie Gao, Ruiqi Cheng, Baocheng Huang

**Affiliations:** 1School of Electronic and Information Engineering, Beihang University, Beijing 100191, China; zhouyang950620@163.com (Y.Z.); ghangzai@163.com (Z.M.); 2Beijing Engineering Research Center of High-Reliability IC with Power Industrial Grade, Beijing Smart-Chip Microelectronics Technology Co., Ltd., Beijing 102299, China; zhaoyang@sgchip.sgcc.com.cn (Y.Z.); gaojie1@sgchip.sgcc.com.cn (J.G.); chengruiqi@sgchip.sgcc.com.cn (R.C.); huangbaocheng@sgchip.sgcc.com.cn (B.H.)

**Keywords:** miniaturized, ultrawideband, magnetic, active probe, high sensitivity, near-field measurement

## Abstract

This article presents a miniature ultrawideband active magnetic probe which is composed of a passive structure and an active amplification circuit structure. The active circuit mainly contains two chips, specifically an amplification chip (HMC797APM5E) and a power management chip (HMC980LP4E). The maximum size of the probe is no more than 64 × 41.5 mm^2^. Compared with the passive probe with the same-sized loop, the sensitivity of the proposed probe is enhanced by 25 dB through the active circuit design. The working frequency bandwidth of the proposed probe can cover 9 kHz to 18 GHz. Additionally, the flatness is about ±4 dB in terms of |S_21_| in the stable working bandwidth. It is efficient for high-frequency near-field scanning.

## 1. Introduction

As high-frequency technology develops, the related problems are becoming more and more serious. The rising complexity of the circuit design and the higher-speed signals cause many electromagnetic compatibility (EMC) issues [1,2,3,4,5]. A lot of unexpected electromagnetic noise is radiated and picked up by the susceptible components, resulting in the instability of the system. So, it is necessary to locate the noise source. However, the complicated pins and the small packages increase the difficulty of the noise location.

A near-field probe can efficiently solve this problem through point-by-point scanning on the surface of the device under test (DUT) [6,7,8]. Traditionally, passive probes have been able to pick up most noise signals. However, when the signals are very week, passive probes cannot detect them. So, many kinds of active probes have been proposed to satisfy this need. Various kinds of active electric or magnetic probes have been developed as a solution for this need [9,10,11]. However, those probes cannot work in a wide bandwidth. The article [12] proposed an active contact current probe, but it cannot measure non-contact magnetic signal. The passive magnetic probe in [13] can work from 9 kHz to 20 GHz, but its transfer factor is only up to −30 dB. The article [14] proposed an ultrawideband active electric probe, which can cover 9 kHz to 18 GHz and has a very high transmission factor. However, research into ultrawideband active magnetic probes is still scarce, because there is a fundamental theoretical difference between magnetic probes and electric probes, and they cannot be substituted for each other in practical application scenarios. Moreover, there have been no successful attempts to integrate the design of ultrawideband magnetic probes with active circuits. For the magnetic probe designed in this article, the front probe tip is a hollow metal loop, and the passive structure is designed with an asymmetric structure. As for the electric probe in [14], the front probe tip is a solid metal structure, and the passive part is designed symmetrically. Moreover, the active structure in this article can provide a stable 20 dB gain, which is better than the 15 dB gain in [14].

Our goal is to design an active magnetic field probe with a high working bandwidth and high transfer factor. Moreover, we will verify the performance of the probe through various standard tests, as well as application tests to demonstrate the ability of the probe to work for a long time. Finally, after theoretical analysis, design, simulation and measurement, we obtain an active magnetic probe which has the characteristics of miniaturization, ultrawide working bandwidth, high sensitivity and high spatial resolution.

## 2. Probe Design Method and Construction

### 2.1. Probe Design Overview

Traditionally, active probes are designed by connecting a separate amplifier at the output of passive probes. This design increases the overall volume of the probe set and is not conducive to placement in a complex and narrow test environment. The probe proposed in this paper focuses on integrated design; the active amplifier circuit and passive probe are directly integrated into an integrated PCB board. Figure 1 shows the overall process of the probe design. First of all, the design working frequency bandwidth of the active probe and the passive structure is determined by the operating frequency band range of the amplifier chip. After the design of passive structure is completed, the design of the amplifier circuit can be started through its simulation data and chip parameters. Finally, an integrated design is carried out to combine the passive part with the active amplification part. After several iterations of the design and ensuring that the simulation results meet the design requirements, we will verify the performance of the design through a series of tests.

The working frequency bandwidth of the existing commercial active magnetic field probes is generally at the 10 GHz level. Additionally, the bandwidth of the academic active magnetic probe is no more than 10 GHz. Moreover, the design method of active probes is very different from that of passive probes. For the design of active probes, the transfer factor and working bandwidth of passive structure, the gain and working bandwidth of amplifier chip should be considered simultaneously. It is worth adding that the transfer factor mentioned in this article is the |S_21_| parameter, which can be expressed by Formula (1), and it can be simply considered as the ratio of output to input power. However, most of the time, it is directly obtained through simulation and testing. At the same time, the transfer factor of a passive probe’s structure with high working bandwidth is generally poor, and the working stability of a high-frequency amplifier chip is also relatively poor. Therefore, if we choose a design with a high working bandwidth, the transfer factor may be sacrificed. So, we chose a more appropriate frequency band as the design goal. Finally, considering the working band of the amplifier chip and the effective working bandwidth of the passive structure, we decided to take 20 GHz as the design goal.
(1)$$\left|S_{21}\right|=\left|\frac{P_{out}}{P_{in}}\right|$$

The overall transfer factor of the probe can be represented by Formula (2). TF_negetive_ represents the transfer factor of the passive structure, which is obtained through simulation. Gain is the gain value of the amplification chip, which can be found through the chip manual. P_loss_ is the loss of active structures. TF_probe_ is the transfer factor of the entire probe, which is the final indicator we use to determine the performance of the probe.

When designing the passive structure, we set the working frequency band at about 20 GHz. Therefore, when we choose the amplification chip, we also focus on 20 GHz. At the same time, we also consider the gain of the chip. Finally, considering the above two elements, we chose the HMC797APM5E. Compared with other chips, this chip can take into account both bandwidth (DC to 22 GHz) and gain (15 dB). However, the passive part of the probe should consider both the flatness of the transfer factor and working bandwidth. As the working bandwidth increases, the active part also introduces more attenuation. After considering the gain and working bandwidth, we finally chose 18 GHz as the target instead of 20 GHz.
(2)$$TF_{negetive}+Gain-P_{loss}=TF_{probe}$$

The probe is composed of four-layer PCB. In order to ensure the stability of the 10 GHz-level working frequency, the dielectric layer of the probe is composed of Rogers material. Figure 2 shows the brief structure of the probe. Additionally, Figure 3 shows the thickness of each dielectric layer and metal layer.

Layer 1 and Layer 3 are used for laying signal power lines. Layer 2 and Layer 4 are designed as ground layers that serve as reference planes for signal and power lines. The power management chip, amplification chip, and most passive components are placed on Layer 1. Layer 4 is used to place only a few devices. The overall design idea of the probe partly refers to [14]. However, there are great differences in design details, because the working principles and physical structures of the electric probe and the magnetic probe are different. In some minor design details, such as the diameter, spacing, number of vias and the width of lines, we obtain the best results through many simulation designs. When designing magnetic probes, our main focus is on the following: the area of the metal loop, the width and length of the transmission line, and the thickness of each metal layer. We will use different parameters for design and ultimately observe the |S_21_| value in the simulation results. It is necessary to ensure that the simulation results have a sufficiently wide working bandwidth and a relatively stable transfer factor.

### 2.2. Design of the Passive Part

The passive part is designed in three parts: the connection part, stripline part and metal loop, as shown in Figure 4. The metal loop is used to detect time-varying magnetic signals in space. To enhance the integrity of signal transmission, the stripeline part is mainly constructed with a stripe line. Finally, connection part is used to connect passive and active part. It connects the signal line from Layer 3 of the PCB to Layer 1 mainly through a through-via.

Figure 5 shows the detail of the metal loop. The metal loop is designed with an isolated through-hole array for impedance matching. Additionally, the corner of the loop is cut at 45 degrees to reduce self-capacitance for better performance in signal transmission and working in a wide-frequency bandwidth.

As Figure 5 shows, when the corner is not cut, most of the signals will be reflected at 180 degrees, resulting in great signal delay. This delay is mainly caused by sudden changes in the width of signal line. The larger the line width, the greater the capacitance value introduced by the change in line width. For example, probes often use line widths at the millimeter level, resulting in delays of up to a ps magnitude. This is a large value for the signal. Choosing a 45-degree angle can significantly delay the change in line width and greatly reduce signal delay.

As a general rule, the performance of the passive magnetic probe is related to the metal loop’s area [13]. Especially for working at high frequency, the area should be smaller compared to working at low frequency. Generally speaking, we first attempt to simulate different size parameters based on some commonly used magnetic field metal loop structures, mainly by adjusting the shape, width, and area of the loop. On this basis, we will optimize some details, such as cutting a 45-degree corner. Finally, we will separately simulate the transfer factor of the metal loop to ensure that it has sufficiently high transmission performance and flatness. So, for the performance at a high frequency, the area of the proposed probe metal loop is set to 16.4 mm^2^. In order to ensure the stability and integrity of signal transmission, the whole passive structure must be designed with impedance matching.

This passive structure can be equivalent, as Figure 6 shows. L_b_ is the equivalent inductance of the DUT (device under test), L_s_ is the equivalent inductance of the metal loop, and C_g_ represents the equivalent capacitance between the metal loop and the ground. The mutual inductance Lm between L_b_ and L_S_ generates current, which is finally amplified by the amplifier circuit and output through the SMA connector. However, this is only a simple model in principle, and the actual equivalent circuit of the probe is much more complex than this. We do not verify the performance of the probe by simulating these equivalent parameters during design. We have more intuitive parameters as reference standards for probe design.

In order to check the actual performance of the passive part, we also fabricated passive structures and obtained some test results for verification. Figure 7 shows the |S_21_| measurement and simulation results of passive structure. As can be seen, the probe working frequency ranges from 9 kHz to 20 GHz. However, starting at 16 GHz, the actual test results are somewhat different.

The gap between |S_21_| simulation and measurement is shown in Figure 8. As can be seen, in the full working frequency range, the transfer factor of the simulation results is greater than the measurement results. The gap is largest in the high-frequency bands. This is also the reason why we chose 18 GHz as the design endpoint, because after 18 GHz, the transfer factor will drop down quickly.

### 2.3. Design of the Circuit Part

Figure 9 shows the brief PCB structure’s top-view of the circuit part. According to the view of Layer 1, it can be divided into four parts: the power part, which is used to connect the external power supply; the output part, which is used to connect SMA connector; the power management circuit; and the amplification circuit. The power management chip uses external power supply to generate appropriate voltage signals for the amplifier chip. In order to ensure the filtering performance of the circuit, capacitors with different values are connected around the chip to avoid the resonance of different frequencies.

Both sides of the signal line in the active circuit are arranged with vias as the signal back flow path. Moreover, the wires connected at the main pins of the chip are matched with filter circuits to reduce noise. Additionally, we keep a certain distance between lines, vias, and cu shapes to avoid mutual interference. Through the above design, the interference generated by the active circuit itself can be reduced to the greatest extent. The input voltage of power management chip is 11.2 V and 3 V.

To check the efficiency of the circuit part, we firstly use circuit-design or ADS to simulate the |S_21_| parameter of the amplification chip. Then, we use the 3dlayout and circuit-design functions in ANSYS to simulate the circuit part model. Figure 10 compares the transfer factor of the circuit part and amplifier chip. Due to the limitation of passive probe itself, its |S_21_| in low-frequency band needs to go through a rising process. When the active part enters the stable working frequency band, the |S_21_| difference compared to the amplification chip is not great. It can be seen that the design of the active part meets our expectations.

### 2.4. Integrated Design

After ensuring that the design of passive and circuit parts meets the requirements, the next step was integrating these parts into a complete active probe. We chose the output signal line at the connection part of the passive part and the input line of the circuit part as the center, then connect these two parts. Figure 2 shows the related model. We used the 3dlayout and circuit-design functions in ANSYS 2020 software to simulate the |S_21_| parameter. In the simulation process, we divided the active probe into two parts after metal-loop part for simulation. The simulation circuit is shown in Figure 11. An impedance of 50 Ω needs to be connected to each of these two parts.

Figure 12 shows a comparison of the |S_21_| results between the active probe and the passive part. The working frequency bandwidth of the active probe ranges from 9 kHz to 18 GHz, which can stably provide a gain of about 20 dB compared with the passive structure. The amplification chip can only provide a theoretical gain of 15 dB. The additional gain of 5 dB may be due to the overall loss of the probe being reduced due to the integrated design.

## 3. Measurement and Validation

This section will introduce the test results and test process of the main performance indicators of the proposed probe. At the same time, we will check the working capability of the proposed probe by scanning a problematic circuit board as an example.

### 3.1. Size Measurement

The size measurement is used to demonstrate the miniaturization of the probe, and to verify its flexibility in a complex testing environment. The specific test process is shown as follows. Firstly, we open the vernier caliper for preheating, and then adjust the measurement result to zero and choose the measurement unit. Then, we measure the dimensions of the probe. Each part should be measured 5 times, and we record the average value as the final parameter. Figure 13 shows the physical picture of the active probe top view, and we add size markings for the main parts.

### 3.2. Frequency Bandwidth and Transfer Factor

Figure 14 shows the physical photo of the measurement system, necessary equipment includes scanner, voltage source, signal generator, microstrip line and proposed probe. In addition, a signal generator is required to provide an input signal to the microstrip line. We connect the end of the microstrip line to signal generator signal input port, and then connect the other end of the microstrip line to a 50 Ohm load as the load end. The voltage source provides two voltages (11.2 V and 3.3 V) to the probe, respectively. Finally, we connect the output port of the probe to the vector network analyzer through an SMA connector. At the same time, the probe needs to be fixed to the robotic arm of the scanner, keeping a distance of 0.1 mm above the microstrip line. When conducting magnetic field testing, it is necessary to minimize the impact of the electric field as much as possible. Generally speaking, during testing, we can determine the maximum magnetic field intensity at that location by continuously rotating the probe. When testing microstrip lines, which have regularity and directionality, the direction of the maximum magnetic field is easily determined through testing and theoretical deduction. It is important to note that the long edge of the probe tip needs to be parallel to the microstrip line. At this angle, the direction of most magnetic field lines is perpendicular to the cross-sectional area of the metal loop, and most of the magnetic field will pass through the metal loop, minimizing the influence of the electric field component.

The measurement process is as follows. Firstly, we open the vector network analyzer and signal generator. Then, we connect the devices according to the above connection requirements. We adjust the parameter settings of the vector network and calibrate it. We set the test parameter as |S_21_| and the measurement bandwidth as 9 kHZ–18 GHz. The output signal of the signal generator and reference amplitude are set to 0 dBm. After checking the connection status and settings, we turn on the voltage source for the power supply. We control the position of the probe through scanner, adjusted with a step of 0.01, and scanned parallel to the center of the microstrip line. When |S_21_| curve reaches a sufficiently large and stable value, we stop scanning and record the image.

Figure 15 shows the simulation results of the active probe and passive probe, and also the measurement result of the active probe. The consistency between the measurement results and the simulation results of the active probe decreases from 4 GHz onwards. After that, the |S_21_| of the measurement results is higher than that of the simulation results until 18 GHz. Compared to the passive probe, the active probe has a significant improvement in the full working bandwidth, and it can provide a gain of 25 dB on average. The active probe will not miss the weak signal in the actual test with such high performance. The actual gain is much higher than the theoretical gain of 15 dB. This may be because the active structure of the proposed probe optimizes the overall transmission performance of the probe, and the working circuit of the amplification chip is reasonably designed to obtain the optimal working performance. Therefore, the final 25 dB gain is not only from the amplification circuit, but also because the performance of the probe itself is optimized after the active structure is integrated.

### 3.3. Sensitivity Test

The section will introduce the sensitivity test of the proposed probe, and compare it with other probes. We choose two probes for comparison. One is a commercial probe (Langer EMV-Near-Field Probe) and the other is a passive magnetic probe for academic research, presented in [13]. This part of the test requires a power source, a signal generator, a signal analyzer and a microstrip line. Firstly, we connect the test devices as follows. Then, we connect the two ends of the microstrip line to the 50 Ω load and signal generator, respectively. After that, we connect the output end of the probe to the signal analyzer and the voltage input ends to the power source. Finally, the probe is placed 1 mm above the center of the microstrip line, at the same time, the metal loop of the probe and the microstrip line remain at 0 degrees. We check the connection of the device and then start the measurement.

Firstly, we select measurement frequency, adjust the input of the signal generator to the maximum, observe the |S_21_| amplitude on the signal analyzer, and continuously reduce the input until the |S_21_| curve on the signal analyzer disappears in the background noise. We record the amplitude on the signal generator at this time, which is the sensitivity of the probe. We use the same method to measure the values of multiple frequency points.

During the measurement process, the signal analyzer has background noise of around −90 dBm. Table 1 shows the sensitivity measurement results of the proposed probe and the comparison with other probes. It can be seen from the results that the proposed probe has better sensitivity than the other two probes, and it will provide excellent performance when measuring weak signals.

### 3.4. Spatial Resolution Test

Ref. [15] describes the definition of spatial resolution (SR). In general, we defined SR as the distance between the −6 dB point and the peak point. The measurement results of SR are affected by the measurement frequency and the distance between the probe and DUT. At the same time, Ref. [16] mentioned the test process. First, we connect the test equipment, fix the probe above the microstrip line, and fix the metal loop of the probe parallel to the microstrip line. We use the scanner to control the probe to scan with a step of 0.1 mm in the direction perpendicular to the microstrip line. We find the maximum point of the output signal amplitude of the probe and stop scanning. This position is the peak point. Then, we take this point as the origin and scan the probe at a distance of −5 mm to 5 mm with a step of 0.5 mm in the direction perpendicular to the microstrip line. We record the output signal amplitude and spatial position of each scanning point, and finally perform normalization processing.

Figure 16 shows the test results of SR. We selected two measurement heights of 0.5 mm and 1 mm, and six test frequencies of 500 MHz, 1 GHz, 2 GHz, 4 GHz, 6 GHz and 8 GHz. It can be seen from the test results that the SR is inversely proportional to the height and directly proportional to the test frequency. When working at 8 GHz, the SR is 0.65 mm (H = 0.5 mm) and 0.7 mm (H = 1 mm). When working at 500 MHz, its SR is 1.3 mm (H = 0.5 mm) and 1.5 mm (H = 1 mm). The probe will provide extremely high accuracy during testing.

### 3.5. Linearity Test

The linearity characteristic is an important technical indicator for active probe. When the input of the signal increases to a certain extent, the output results of the active probe will have a nonlinear distortion. According to the amplification chip manual shown in Table 2 and test results, the proposed active probe can withstand up to 27 dBm input power. Otherwise, the amplifying circuit could be damaged due to the over-voltage. It can be seen from the following parameters that the probe has a certain immunity in the face of sudden changes in current and voltage.

### 3.6. Applications in Near-Field Measurement

In this section, we select a problem circuit board with certain noise radiation and a microstrip line as the test objects to check the working performance of the probe. The main part is a clock management chip (9LPRS419DFLF) and its bypass circuit. The chip radiates a lot of electromagnetic waves, at a higher level than the surrounding circuits do.

The test area is limited in the red grid lines and occupies an area of 20 cm by 40 cm. After the probe, the signal analyzer and scanner are connected in a suitable manner. We place the probe 0.5 mm above the circuit board. The movement of probe is controlled by a scanner described in [17]. Then, the probe moves 0.1 mm at a time and then record the test result. When the scanner completes the whole progress, we turn the individual data into a radiation graph. Additionally, a passive magnetic probe in [18] is chosen for comparison.

When the circuit board is supplied with power, it will emit electromagnetic radiation in all frequencies. First, we manually adjust the test frequency of the signal analyzer to find a frequency value with larger amplitude of the test signal. Then, we use the probes to scan over the red-line squares and record the amplitudes. Finally, we use 500 MHz as the target frequency.

Figure 17 shows the measurement results, indicating that the passive probe can only detect the a −73.34 dBm signal, while the proposed active probe can detect a −58.63 dBm signal. At the same time, by comparing the relationship between the corresponding colors of the signals measured by each probe in the two images, it is found that the amplitude of the signal captured by the active probe is greater than that of the passive probe. This suggests that the active probe can provide more stable gain in the whole scanning region. Overall, this means that the active probe can capture radiation signal with less loss. The whole scanning process took more than 30 h, which shows that the proposed active probe can work stably for a long time and provide very accurate test results.

In order to verify the test capability of the probe in the GHz band, we selected a microstrip line as the test object. We used a signal generator to provide a 3 GHz signal with an amplitude of −10 dBm for this microstrip line. The overall scanning range was 5 cm × 9 cm. The scanning step is 0.2 mm. The overall scanning time was over 40 h. Figure 18 shows the scanning results. Similarly, we can see that in the scanning results of the active magnetic field probe, the maximum value (−19.41 dBm) of signal amplitude is larger than passive magnetic field probe (−34.52 dBm). From the comparison result, it can be seen that regarding GHz level, in the scanning test on the DUT, the probe proposed in this paper has better performance.

Table 3 shows the comparison of key parameters among several probes. It can be seen that the probe proposed in this article has certain advantages in various aspects.

## 4. Conclusions

To obtain high performance in near-field scanning, an active magnetic probe is proposed in this paper. The proposed probe has obvious advantages in several parameters. Firstly, the proposed probe can work in a wider working bandwidth, and provides high gain. At the same time, the probe size in this paper is small enough, and the spatial resolution is also at a good level. It is worth noting that commercial active probes are usually designed in a separated manner. The integrated design of the proposed probe efficiently reduces the size of the entire probe compared to the separated design, which makes measurement more convenient.

The active amplification circuit, passive construction and the integrated design all have efficient simulation or measurement results. All kinds of indexes have significant improvements compared with the probes for conference. Overall, the active probe provides a gain of about 25 dB over the passive part. It is evident that the integrated design meets expectations. Additionally, the working bandwidth can cover from 9 kHz to 18 GHz. Meanwhile, the size of the probe is no more than 64 × 41.5 mm^2^, and the probe has a high spatial resolution at high frequency. The probe can also handle some unstable test changes. In general, this probe could handle most measurement environments while detecting weak signals in a GHz bandwidth.

## Figures and Tables

**Figure 1 sensors-23-06170-f001:**
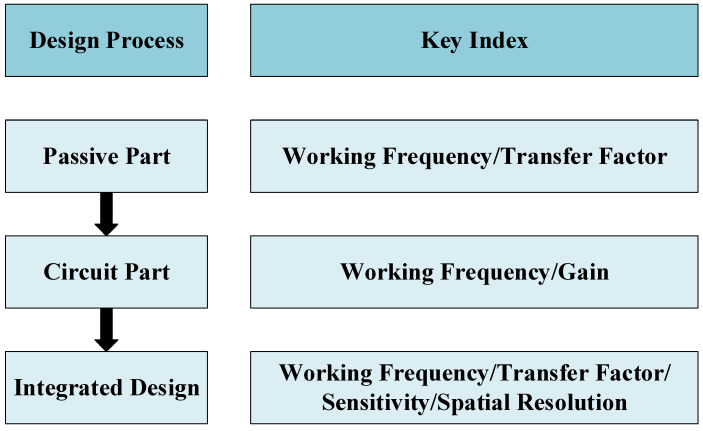
Probe design process.

**Figure 2 sensors-23-06170-f002:**
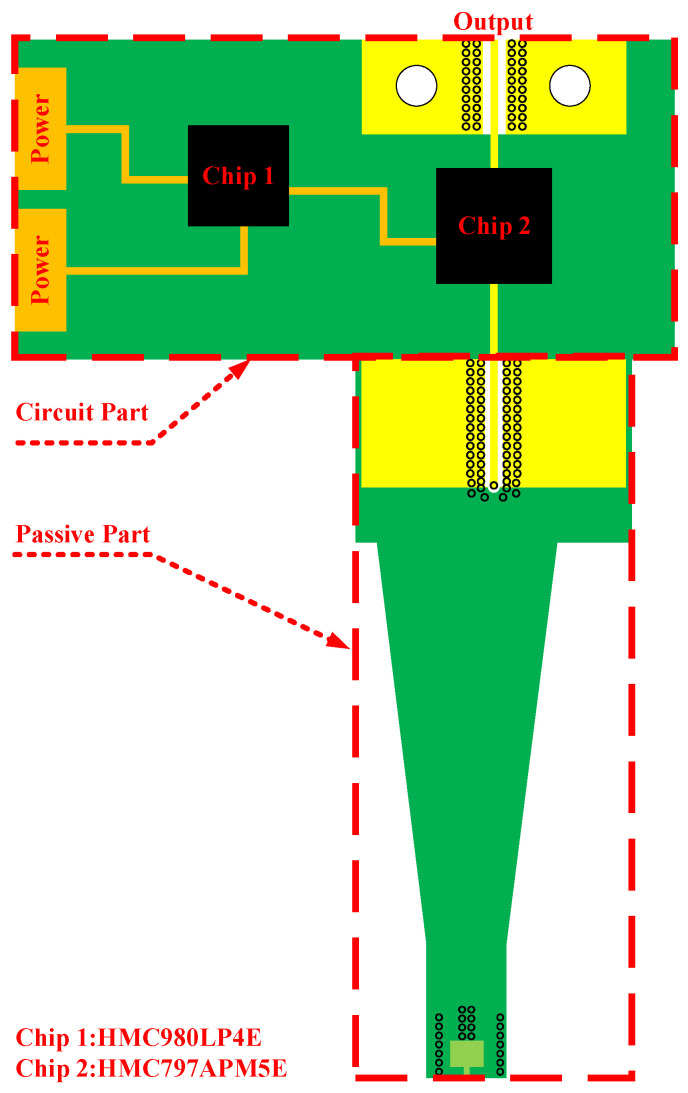
Brief structure of proposed probe.

**Figure 3 sensors-23-06170-f003:**
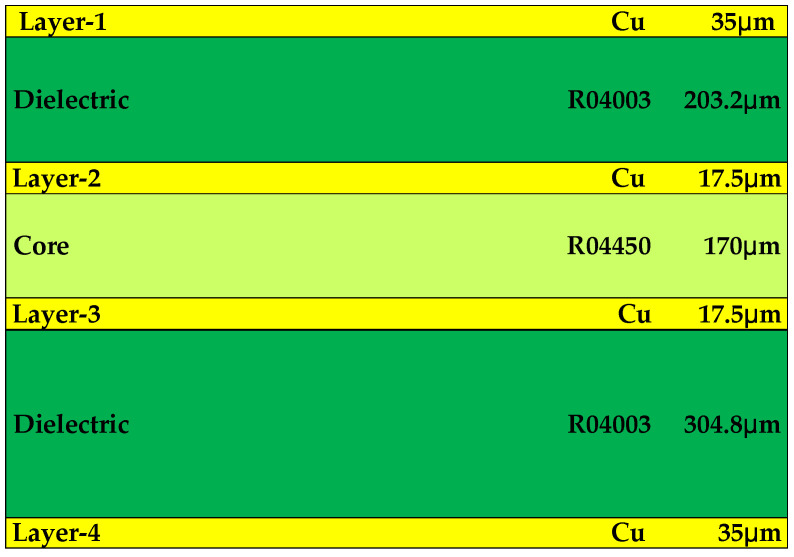
Schematic diagram of probe stack size.

**Figure 4 sensors-23-06170-f004:**
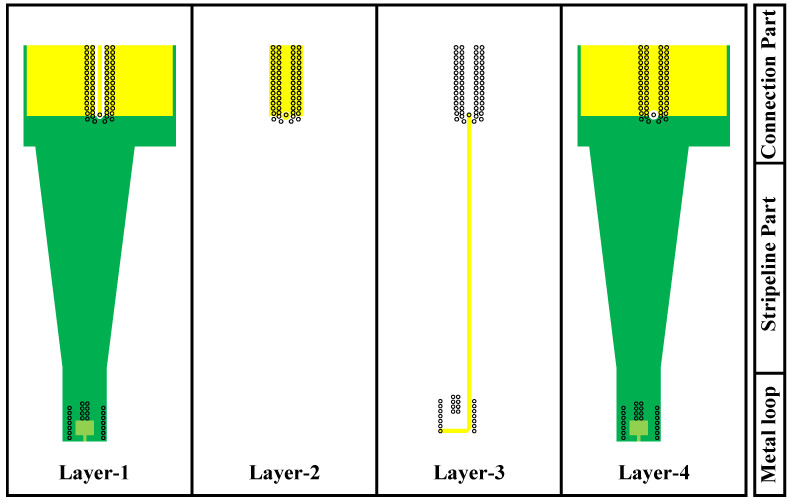
Structure diagram of each layer of the passive part.

**Figure 5 sensors-23-06170-f005:**
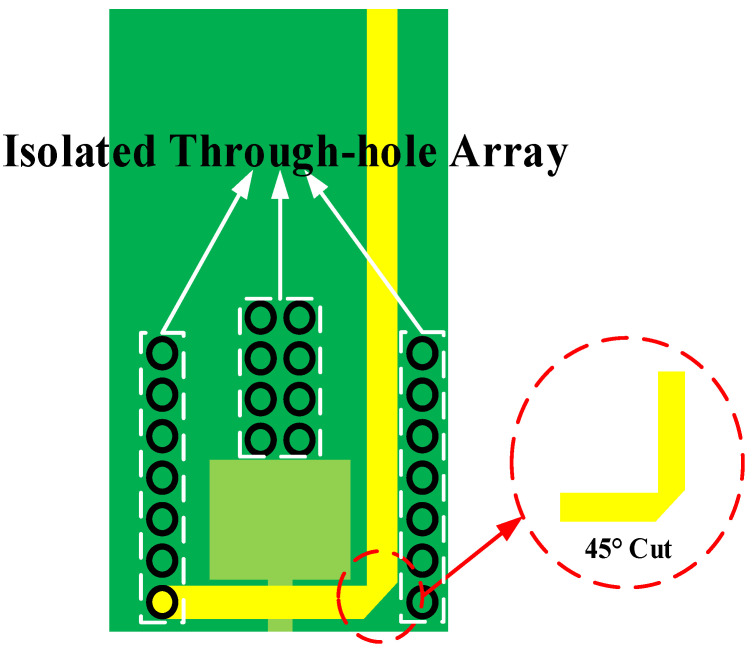
The detailed structure of the metal loop.

**Figure 6 sensors-23-06170-f006:**
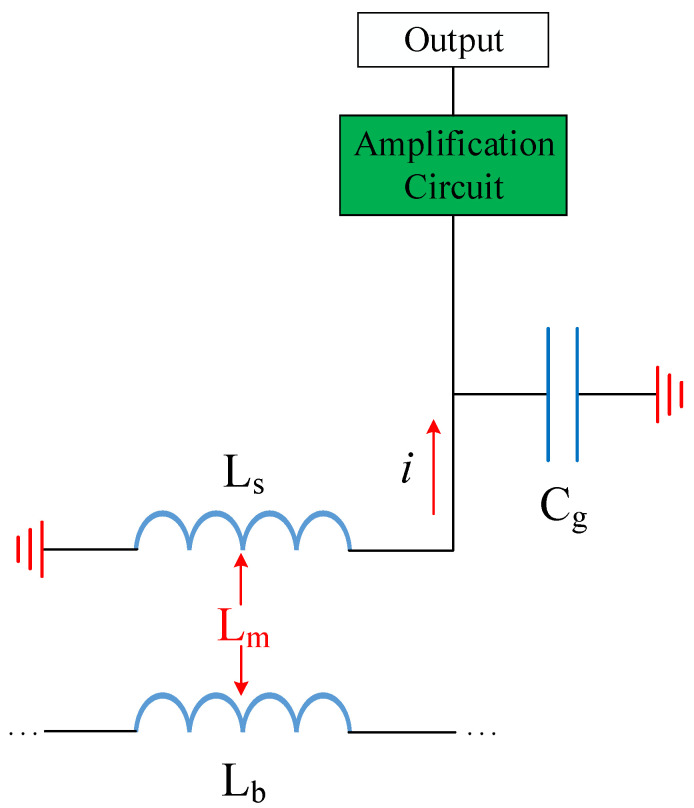
Passive part equivalent circuit model.

**Figure 7 sensors-23-06170-f007:**
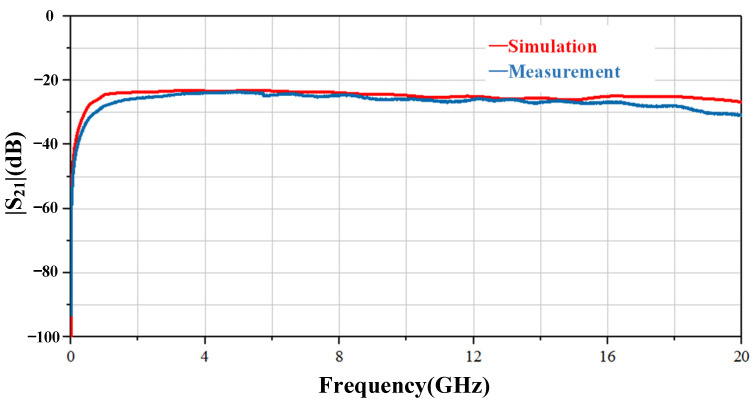
|S_21_| simulation and measurement results of the passive part.

**Figure 8 sensors-23-06170-f008:**
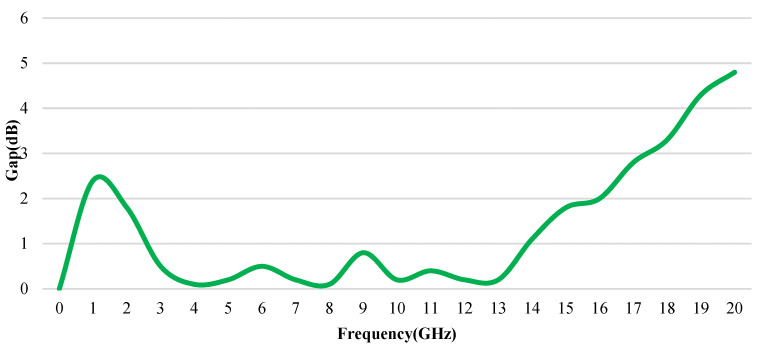
Gap for |S_21_| simulation and measurement.

**Figure 9 sensors-23-06170-f009:**
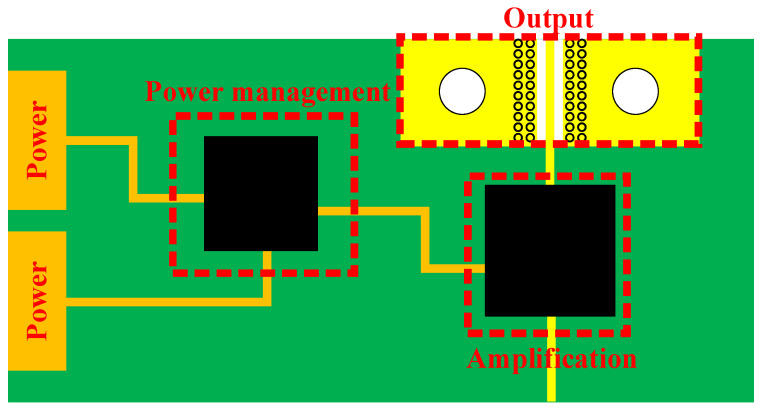
Brief diagram of the circuit part’s Layer 1.

**Figure 10 sensors-23-06170-f010:**
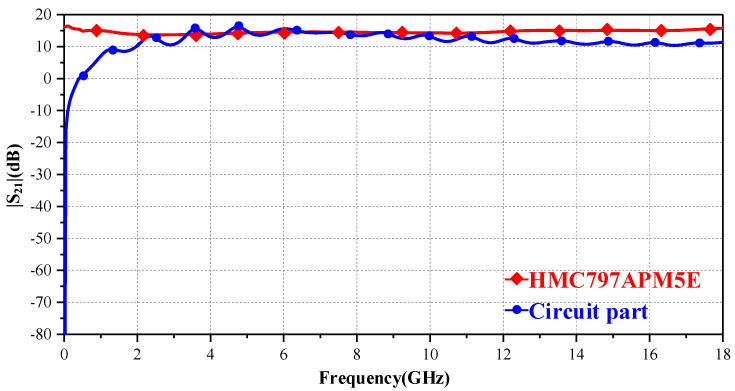
|S_21_| results of the circuit part and amplification chip.

**Figure 11 sensors-23-06170-f011:**
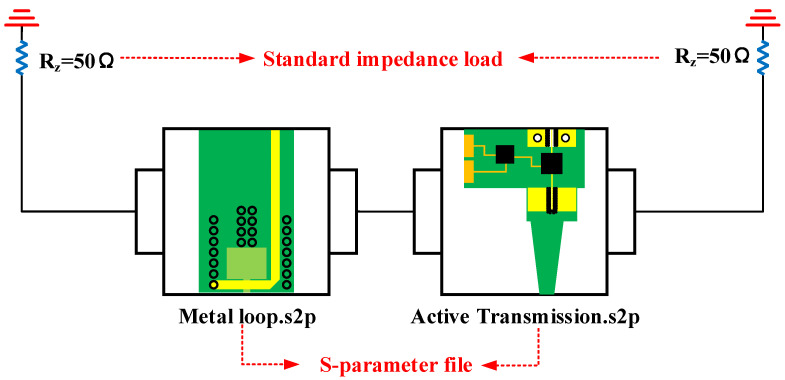
Circuit design simulation model.

**Figure 12 sensors-23-06170-f012:**
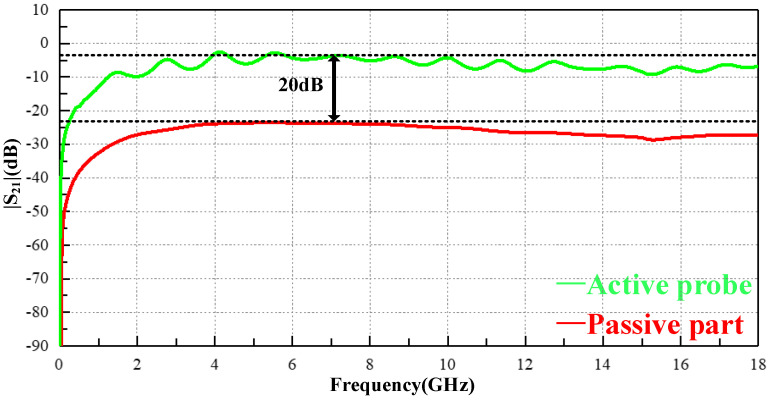
Comparison between the passive part and active probe.

**Figure 13 sensors-23-06170-f013:**
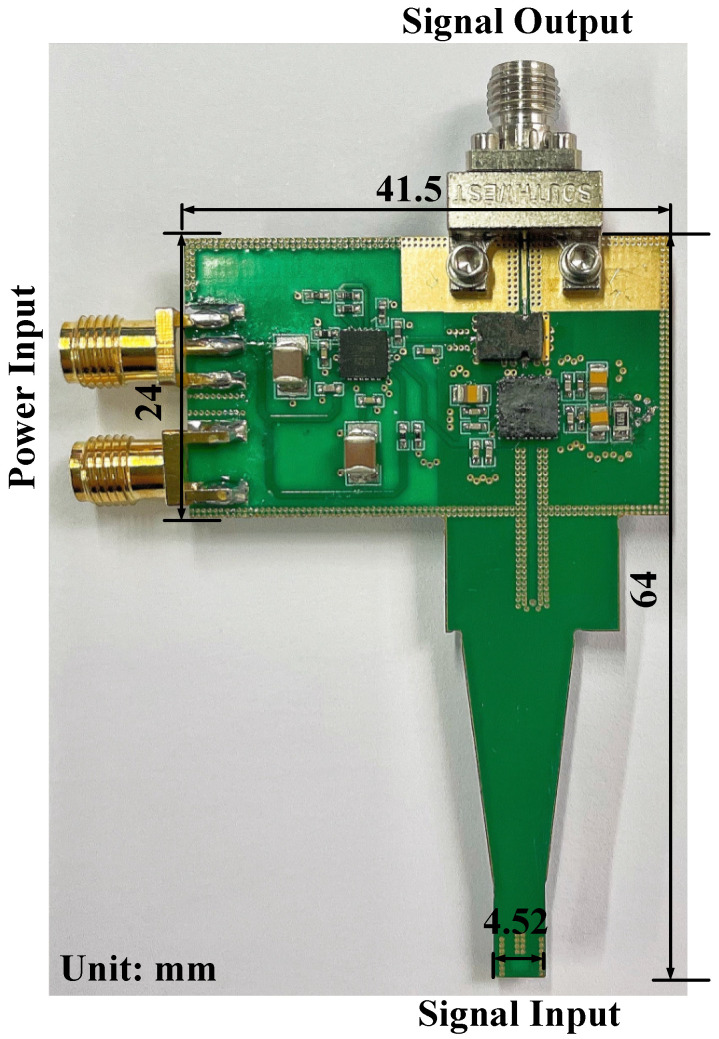
The physical picture of the active probe top view.

**Figure 14 sensors-23-06170-f014:**
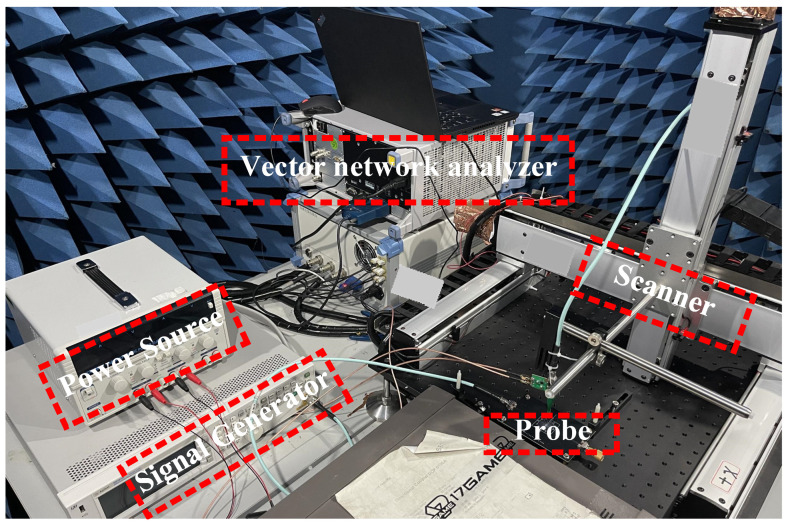
The measurement system environment.

**Figure 15 sensors-23-06170-f015:**
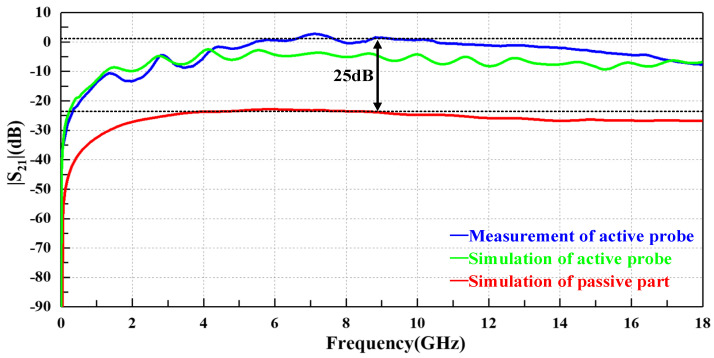
Comparison among the simulation of passive part, measurement and simulation of the active probe.

**Figure 16 sensors-23-06170-f016:**
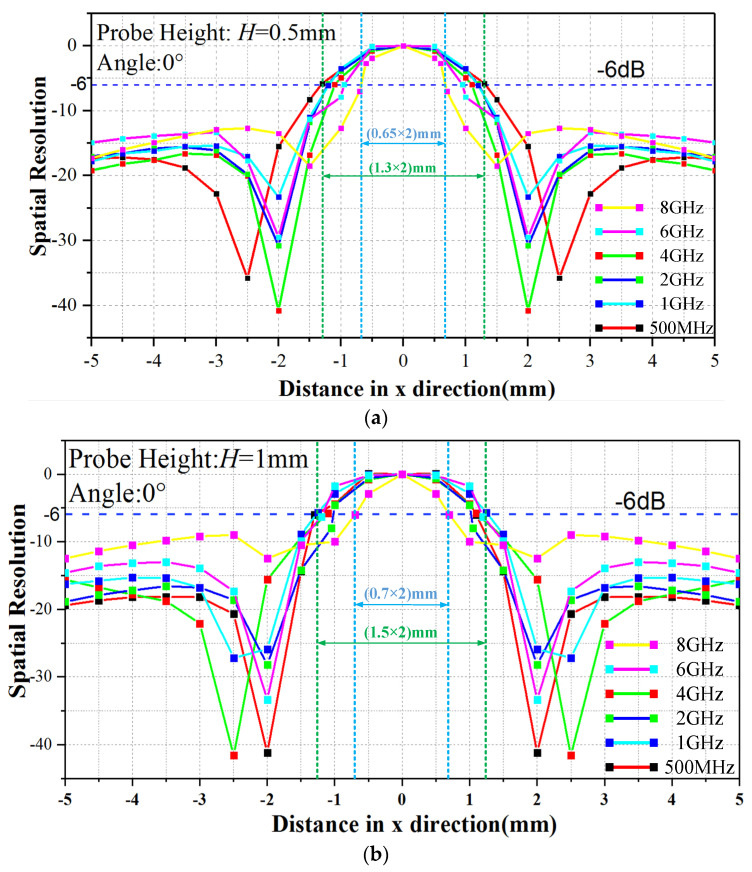
The spatial resolution with different heights (H). (**a**) H = 0.5 mm and (**b**) H = 1 mm.

**Figure 17 sensors-23-06170-f017:**
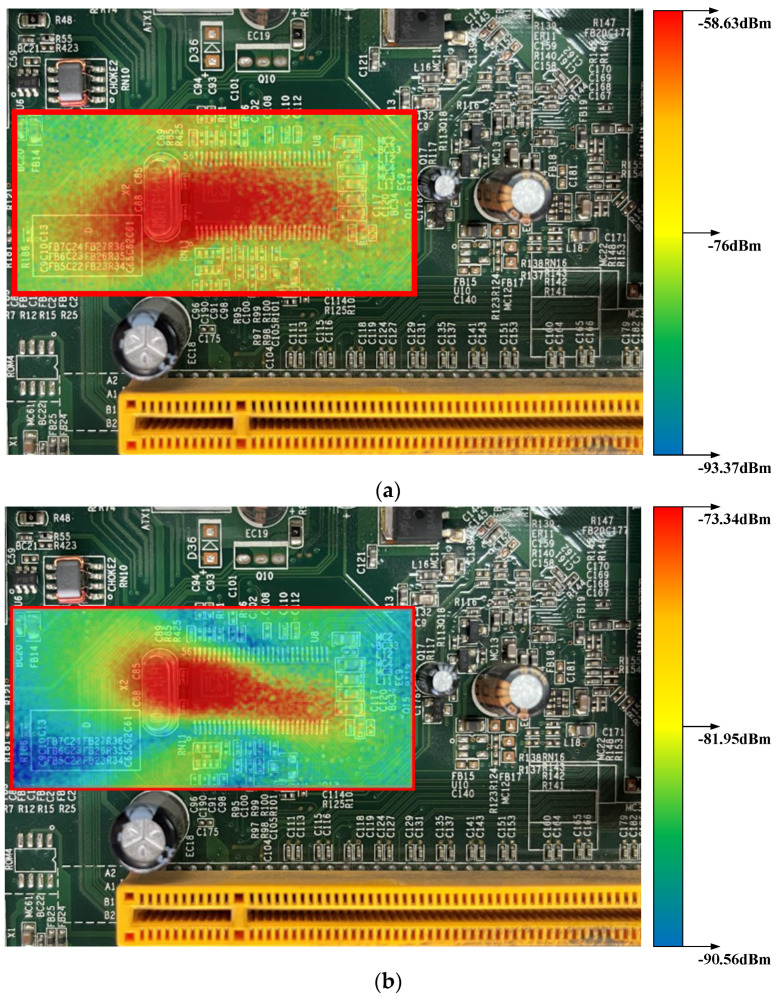
The near-field scanning results of the circuit board: (**a**) proposed active probe; (**b**) passive probe.

**Figure 18 sensors-23-06170-f018:**
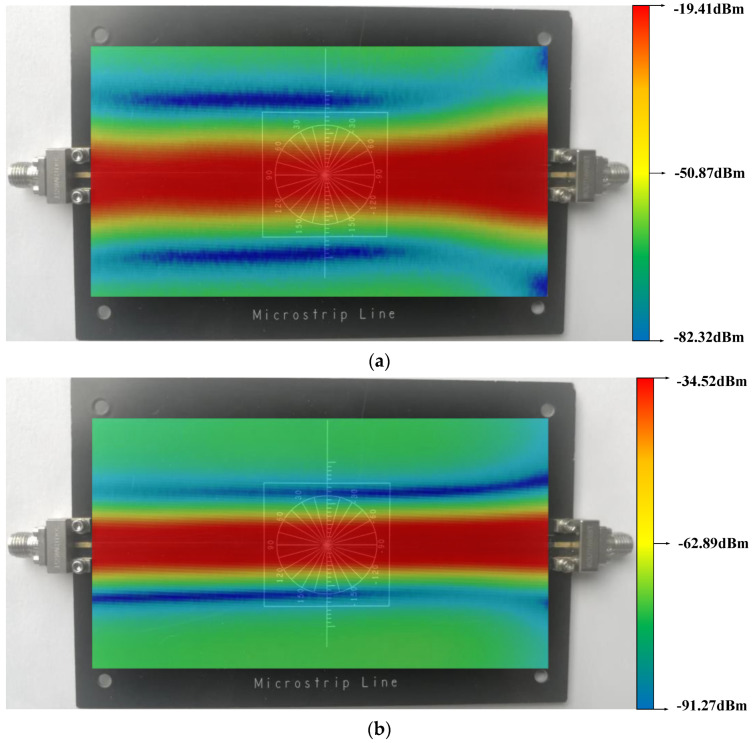
The near-field scanning results of microstrip line: (**a**) proposed active probe; (**b**) passive probe.

**Table 1 sensors-23-06170-t001:** Sensitivity of the proposed probe and comparison.

Frequency	Proposed Probe	Commercial Probe	Probe in [13]
9 kHz	10 dBm	16 dBm	13 dBm
1 MHz	−25 dBm	−15 dBm	−20 dBm
500 MHz	−50 dBm	−35 dBm	−40 dBm
1 GHz	−70 dBm	−51 dBm	−56 dBm
3 GHz	−71 dBm	−52 dBm	−55 dBm
5 GHz	−72 dBm	−57 dBm	−56 dBm
10 GHz	−74 dBm	−20 dBm	−54 dBm
14 GHz	−72 dBm		
18 GHz	−74 dBm		

**Table 2 sensors-23-06170-t002:** Main parameters of the amplifier chip.

Parameter	Value
High P1dB Output Power	29 dBm
High Psat Output Power	31 dBm
High Gain	15 dB
High Output IP3	41 dBm
RF Input Power	27 dBm
Matched Input/Output	50 Ohm
Operating Temperature	−40 to +85 °C
ESD Sensitivity (HBM) Class	1A—Passed 250 V

**Table 3 sensors-23-06170-t003:** Comparison with other active magnetic probes.

Probe	BW (Hz)	Size (mm × mm)	TF (dB)	SR (mm)	Gain (dB)
Proposed Probe	9 k~18 G	64 × 41.5	<0	0.65	25
Probe in [10]	9 k~1 G	60 × 19	<0	0.9	33
Commercial Probe	1 M~6 G	-	-	0.3	-
Probe in [13]	9 k~20 G	85 × 15	<−30	-	-

BW: Bandwidth. TF: Transfer factor. SR: Spatial resolution. Commercial Probe: Langer MFA-R 0.2-75.

## Data Availability

Not applicable.

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
