# Peer review of "Design of Miniature Ultrawideband Active Magnetic Field Probe Using Integrated Design Idea"

_sensors, 2023, doi:10.3390/s23136170_

Round 1
Reviewer 1 Report
This is a nice work.
Please find my comments in the pdf file.

The PDF file indicates a few small problems with the English language.
Author Response
To #Reviewer 1
Thank you very much for your meticulous review and patient response. We have had a lot of discussion and reflection on your opinions, and have finally adjusted them one by one. The following is the response to your question. The modifications made to the following issues have been marked in green in the text.
#Based on the feedback from Reviewer 2, we have made some deletions to the content of the paper, including some of the content that you have commented on. So some of the answers are inconvenient to present in the original text. However, we will still provide corresponding answers in the reply.
- Please compare your work with ref [14] in more details, and justify your novelty over this work.
Response: The active magnetic probe mentioned in this article is fundamentally different from the active electric probe in [14]. Mainly in the design of passive structures. Firstly, the electric probe relies on the solid metal substrate in the front section to induce an electric field, which is then converted into a current signal. However, the magnetic probe relies on a hollow ring at the front end to sense the magnetic field signal and convert it into a current signal. The resonance frequency and noise point of the signal will be different, and therefore, when designing the active circuit, the matched filtering circuit design will also be different. Our innovation lies in designing an ultra-wideband passive magnetic probe and integrating it with active circuits.
- Kindly provide the formula for transfer factor.
Response: In fact, we assume that the transfer factor is the S21 parameter which can be expressed by formulas (1), it can be simply considered as the ratio of output to input power.
(1)
- Please try to formulate your analysis rather than explaining in the content.
Response: In fact, the design of the probe has a gradual process. For example, the working bandwidth of the probe is gradually increased from kHz to GHz level. In this process, we will summarize some corresponding experiences. As the working frequency of the probe increases and to ensure the flatness of the transfer factor, the loss on the transmission line will increase. So we need to balance the relationship between the bandwidth and the transfer factor. In our design process, the first goal is to achieve a stable transfer factor close to 0 for the active probe. And considering that the upper limit of the working frequency of existing active magnetic probes is basically below 5~10 GHz, we do not intend to significantly increase the bandwidth at once. The transfer factor of relatively good passive magnetic probes can generally approach -20 dB. So when selecting chips, our goal is to have a working bandwidth of around 20 GHz and a gain value of 15~20dB.
The overall transfer factor of the probe can be represented by formula (2). TFnegetive represents the transfer factor of the passive structure, which is obtained through simulation. Gain is the gain value of the amplification chip, which can be found through the chip manual. Ploss is the loss of active structures. TFprobe is the transfer factor of the entire probe which is the final indicator we use to determine the performance of the probe.
(2)
- Kindly explain your method to achieve the best result. You may include the trend that you followed through your simulation results to reach to the best result.
Response: When designing magnetic probes, our main focus is on the following: the area of the metal loop, the width and length of the transmission line, and the thickness of each metal layer. We will use different parameters for design and ultimately observe the S21 value in the simulation results. It is necessary to ensure that the simulation results have a sufficiently wide working bandwidth and a relatively stable transfer factor.
- Kindly quantify the loss in both cases.
Response: The loss mentioned here should be more suitable to be referred to as delay. The delay is mainly caused by sudden changes in the width of signal line. The larger the line width, the greater the capacitance value introduced by the change in line width. For example, probes often use line widths at the millimeter level, resulting in delays up to ps magnitude. This is a large value for the signal. Choosing a 45 degree angle can significantly delay the change in line width and greatly reduce signal delay.
- Kindly explain how the metal loop area was chosen.
Response: Similar to the answer in question 4, we have multiple considerations when determining the area of the metal loop. For example, it is necessary to ensure that the metal loop is as close as possible to the tested equipment. This allows more magnetic field lines to pass through the metal loop. At the same time, the area of the metal loop should not be too large, otherwise it will reduce the induced current value. But it should not be too small, otherwise the magnetic field lines passing through the metal ring will decrease. When designing the size and mechanism of the metal loop, there are actually many different designs, each with its own advantages and disadvantages. Ultimately, it is necessary to judge whether the current results are acceptable through simulation. The structure presented in the article only achieves the best balance between different strengths and weaknesses.
- Kindly provide the values in the equivalent circuit model.
Response: In fact, this is just a physical model, and their actual values are not calculated during design. It is only used to demonstrate the working principle of the probe. Because the actual equivalent physical model of the probe will be much more complex than in the figure, calculating them will not help improve the efficiency of the design. We have more intuitive parameters as reference standards for probe design. For example, the S21 parameter in simulation results.
- Please pay attention to subscripts: Lb, Ls, Cg
- Kindly include the version.
- Kindly revise the English.
- Kindly revise the English.
- Kindly revise the English.
- Please correct the citation style in this table.
Response: For questions 8 to 13, we have made modifications in the areas you marked in the text and mark then in green.

Reviewer 2 Report
The manuscript presents a study of the design of a miniature ultrawideband magnetic field probe. In going over the manuscript, I note that the authors need to address the following:
1. Magnetic fields are extremely direction-dependent; how is this aspect of magnetic fields addressed in the present study?
2. I would have liked to see some studies of orientation-dependence of the magnetic field measurements in the manuscript;
3. It appears that several figures - Figures 2-8, 11, 12 can be included in Supplementary Materials; for a manuscript of this length, there are too many figures;
4. What are the error bars in Figures 9 and 10?
In general, the language can be improved throughout the paper.
On a personal note, I am extremely sorry for the delay in the submission of my review of the manuscript.
The manuscript presents a study of the design of a miniature ultrawideband magnetic field probe. In going over the manuscript, I note that the authors need to address the following:
1. Magnetic fields are extremely direction-dependent; how is this aspect of magnetic fields addressed in the present study?
2. I would have liked to see some studies of orientation-dependence of the magnetic field measurements in the manuscript;
3. It appears that several figures - Figures 2-8, 11, 12 can be included in Supplementary Materials; for a manuscript of this length, there are too many figures;
4. What are the error bars in Figures 9 and 10?
In general, the language can be improved throughout the paper.
On a personal note, I am extremely sorry for the delay in the submission of my review of the manuscript.
Author Response
To #Reviewer 2
Thank you very much for your meticulous examination of this article and constructive advice. In response to your feedback, we have made the following response and hope to answer your question. The modifications made to the following issues have been marked in yellow in the text.
- Magnetic fields are extremely direction-dependent; how is this aspect of magnetic fields addressed in the present study?
Response: Taking microstrip lines as an example, we usually assume that the magnetic field revolves around the direction of the line. For complex and dense circuits or chips, we usually rotate the magnetic probe continuously to capture the maximum value of the received signal. At this point, the magnetic field will be concentrated in the direction perpendicular to the probe metal loop.
- I would have liked to see some studies of orientation-dependence of the magnetic field measurements in the manuscript.
Response: Generally speaking, when using the magnetic probe for testing, there will be interference from some electric field components, so it is necessary to ensure that the electric field signal captured by the magnetic probe is as small as possible. When conducting scanning tests, as the test object is constantly changing and usually not regular, we do not determine the maximum magnetic field value at each point by rotating the probe. But scanning all objects at a fixed probe angle, then check the maximum magnetic field value of special points. When the test object is a fixed position or a regular structure like microstrip line, it is necessary to determine the maximum value of the magnetic field by rotating the probe.
- It appears that several figures - Figures 2-8, 11, 12 can be included in Supplementary Materials; for a manuscript of this length, there are too many figures.
Response: We have made deletions to some of the graphics mentioned above, and also processed the relevant text parts.
- What are the error bars in Figures 9 and 10?
Response: We add Figure 8 to illustrate the gap between simulation and measurement of passive structures.As can be seen, In the full working frequency range, the transfer factor of the simulation results is greater than the measurement results. And the gap comes to largest in the high frequency bands. This is also the reason why we chose 18 GHz as the design endpoint, because after 18 GHz, the transfer factor will drop down quickly.
Figure 8. Gap for |S21| simulation and measurement.
